

# Seasonal changes in the abundance and biomass of copepods in the south-eastern Baltic Sea in 2010 and 2011

Lidia Dzierzbicka-Glowacka[1,*], Anna Lemieszek[2,*], Marcin Kalarus[2,*] and Evelina Griniene[3,*]

[1] Physical Oceanography Department, Ecohydrodynamics Laboratory, Institute of Oceanology of the Polish Academy of Sciences, Sopot, Poland
[2] Department of Ecology, Maritime Institute in Gdansk, Gdansk, Poland
[3] Marine Research Institute, Klaipeda University, Klaipeda, Lithuania
[*] These authors contributed equally to this work.

Corresponding author
Lidia Dzierzbicka-Glowacka,
dzierzb@iopan.gda.pl

## ABSTRACT

**Background**. Copepods are major secondary producers in the World Ocean. They represent an important link between phytoplankton, microzooplankton and higher trophic levels such as fish. They are an important source of food for many fish species but also a significant producer of detritus. In the terms of the role they play in the marine food web, it is important to know how environmental variability affects the population of copepods.

**Methods**. The study of the zooplankton community in the south-eastern Baltic Sea conducted during a 24-month survey (from January 2010 to November 2011) resulted in the identification of 24 invertebrate species (10 copepods, seven cladocerans, four rotifers, one ctenophore, one larvacean, and one amphipod). Data were collected at two stations located in the open sea waters of the Gulf of Gdansk: the Gdansk Deep (P1) (54°50′N, 19°19′E) and in the western, inner part of the Gulf of Gdansk (P2) (54°32′N, 18°48.2′E). The vertical hauls were carried out with the use of two kinds of plankton nets with a mesh size of 100 $\mu$m: a Copenhagen net (in 2010), and a WP-2 net (in 2011).

**Results**. The paper describes the seasonal changes in the abundance and biomass of copepods, taking into account the main Baltic calanoid copepod taxa (*Acartia* spp., *Temora longicornis* and *Pseudocalanus* sp.). They have usually represented the main component of zooplankton. The average number of copepods at the P1 Station during the study period of 2010 was 3,913 ind m$^{-3}$ (SD 2,572) and their number ranged from 1,184 ind m$^{-3}$ (in winter) to 6,293 ind m$^{-3}$ (in spring). One year later, the average count of copepods was higher, at 11,723 ind m$^{-3}$ (SD 6,980), and it ranged from 2,351 ind m$^{-3}$ (in winter) to 18,307 ind m$^{-3}$ (in summer). Their average count at P2 Station in 2010 was 29,141 ind m$^{-3}$, ranging from 3,330 ind m$^{-3}$ (in March) to 67,789 ind m$^{-3}$ (in May). The average count of copepods in 2011 was much lower at 17,883 ind m$^{-3}$, and it ranged from 1,360 ind m$^{-3}$ (in April) to 39,559 ind m$^{-3}$ (in May).

**Discussion**. The environmental conditions of the pelagic habitat change in terms of both depth and distance from the shore. Although the qualitative (taxonomic) structure of zooplankton is almost identical to that of the coastal waters, the quantitative structure (abundance and biomass) changes quite significantly. The maximum values of zooplankton abundance and biomass were observed in the summer season, both in the Gdansk Deep and in the inner part of the Gulf of Gdansk. Copepods dominated

in the composition of zooplankton for almost the entire time of the research duration. Quantitative composition of copepods at the P1 Station differed from the one at P2 Station due to the high abundance of *Pseudocalanus* sp. which prefers colder, more saline waters.

## INTRODUCTION

Zooplankton in the marine pelagic food webs plays an important role in the energy transfer between primary producers (phytoplankton) and higher-level consumers, like pelagic fish (*Möllmann, Kornilovs & Sidrevics, 2000*; *Dzierzbicka-Glowacka, Bielecka & Mudrak, 2006*; *Dzierzbicka-Glowacka et al., 2012*; *Dzierzbicka-Glowacka, Kalarus & Zmijewska, 2013*). In terms of biomass and production, copepods are the most important taxa of zooplankton in the southern Baltic Sea, e.g., *Pseudocalanus* sp., *Temora longicornis* and *Acartia* spp., while rotifers are mainly represented by *Synchaeta* spp., and Cladocera by the dominance of *Evadne nordmanni* (*Dzierzbicka-Glowacka et al., 2015*). The species of *Pleurobrachia pileus* belonging to Ctenophora, the copepod *Eurytemora affinis* and rotifers *Keratella* spp. are the least important taxa in the biomass and in the production of zooplankton (*Wiktor, 1990*; *Wiktor & Zmijewska, 1996*; *Mudrak & Zmijewska, 2007*). Between the dominant species and those from the bottom of the scale, there are intermediate species living in the Baltic Sea which are characterized by very similar biomass values, e.g., *Fritillaria borealis* (Appendicularia), the larvae of Polychaeta and Bivalvia, cladocerans *Bosmina* spp. and *Podon* spp., as well as the copepod *Centropages hamatus* (*Andrulewicz et al., 2008*).

The spatial variation in the species composition of mesozooplankton results primarily from the salinity of the Baltic Sea. The lowest number of species (13–20) occurs in the central region of the Baltic Proper and increases in the marine and freshwater regions. The highest number of species (approx. 28–32) is encountered in the south-western part of the Baltic Proper, which is strongly affected by the North Sea (*Andrulewicz et al., 2008*).

Copepods are one of the most important links in the food web. They play an important role in the transmission of energy between producers and consumers of higher orders, being food for many pelagic, planktivorous fish (*Williams, Conway & Hunt, 1994*; *Froneman et al., 1996*). Their abundance is highly dependent on the physicochemical variables of the environment (*Möllmann, Kornilovs & Sidrevics, 2000*; *Möller et al., 2015*; *Karlsson, Puiac & Winder, 2018*).

The main objective of the study was to describe the seasonal changes in the abundance and biomass of the major Baltic copepod species (*Acartia* spp., *T. longicornis*, and *Pseudocalanus* sp.) in the Gdansk Basin (the south-eastern Baltic Sea). The data obtained will be used as a background for future numerical evaluations.

## MATERIAL AND METHODS

### The Study area

The Baltic is a shallow shelf from the group of internal (intracontinental) seas. It is the youngest European sea and one of the youngest seas of the Atlantic Ocean. It covers an area of approx. 415,000 km². It is connected with the North Sea through a number of straits: the Danish Straits (Sund, Little Belt and Great Belt), Kattegat, and Skagerrak. The generally accepted division of the Baltic Sea, based on the seabed topography, enables the identification of regions with clearly defined hydrographic parameters (*Fonselius, 1969*; *Omstedt, 1990*), i.e., the Gulf of Bothnia, the Bothnian Sea, the Gulf of Finland, the Gulf of Riga, the Baltic Proper (the southern Baltic), the Danish Straits, and Kattegat.

The Baltic waters are characterized by fluctuations in salinity resulting from, i.a., irregular inflows of fresh and saline waters. The inflows of saline waters occur in the western part of the sea through the Danish Straits, which connect the Baltic Sea and the North Sea. This phenomenon contributes to the two-layer structure of the Baltic waters (*Matthäus & Franck, 1992*; *Fonselius & Valderrama, 2003*; *Leppäranta & Myrberg, 2009*). The upper layer consists of lighter waters with salinity ranging from 20 PSU in the Kattegat to 2–3 PSU at the northern end of the Gulf of Bothnia and the eastern end of the Gulf of Finland, and with 8 PSU in the Baltic Proper. The surface waters are well-mixed and well-oxygenated, and their temperature differs depending on the season from 0 °C to 20 °C. The lower, deep water zone is characterized by the basically constant temperature of 4–6 °C and higher salinity, ranging from 12 to 20 PSU depending on the region. Stability between these zones is attributed to the halocline, which separates surface waters from the deep water layer, preventing mixing of the waters at the open sea area in particular. The Southern Baltic is an area of particular importance to the entire Baltic Sea. Saline waters from the North Sea pass through this region of the Baltic. The direction of the near-bottom flows is affected by the seabed topography. The Słupsk Furrow, with a maximum depth of 92 m and with a width of 40 km, is a gateway through which the inflow waters move eastwards from the North Sea. Water inflows from the North Sea raise the salinity of the Baltic waters. The oxygen content and the dynamics of the temperature are determined by the seasons. Although the Gdansk Deep is located off the inflow-water transit axis, it plays an important role in this process (*Osiński, 2008*).

The environment of Gdansk Basin is determined by a varying volume of river runoff; the easy exchange of water with the Baltic Sea, including periodical inflows (infusions) of seawater; and highly variable physicochemical conditions.

Seasonal temperature changes occurring in the upper water layer result from seasonal variability in the meteorological elements. They are affected mainly by vertical processes, in particular convection and wind mixing, as well as the Vistula River water inflows into the Gulf of Gdansk, which raise the water temperature in the spring-summer season and lower it in the autumn-winter season (*Cyberski, 1995*). The water temperature in the layer above 80 m gradually increases up to the maximum value at the bottom. Due to a lack of contact with the atmosphere, deep waters do not exhibit seasonal changes typical of
the upper layer, and their temperatures depend on the temperatures of the inflow waters (*Majewski, 1990*).

The distribution of salinity throughout the year in the surface layer of the Gdansk Basin is affected by the varying volume of river waters reaching the Basin and affecting the anemobaric conditions. Salinity shows a clear seasonal variability in the shallow littoral zone. Differences in the vertical stratification of salinity result from the interactions between the Vistula waters, which reduce salinity, and the deep waters, which increase salinity (*Majewski, 1990*). The salinity of benthic waters (above the level of 80 m) also depends on the inflows of saline waters from the North Sea.

## Sampling

Plankton samples, which are the basis of *in situ* studies, were collected in the southern part of the Baltic Sea at the two stations: in the Gdansk Deep (P1) and in the western part of the Gulf of Gdansk (P2).

The first series consists of the biological material collected aboard the r/v "Oceania", from the Institute of Oceanology of the Polish Academy of Sciences, during seven cruises in the area of the Gdansk Deep (54°50′N, 19°19′E) (Fig. 1, P1 Station), in the period from February 2010 to November 2011. The maximum depth of this site is approx. 100 m.

The vertical hauls were carried out with the use of two plankton nets with 100 μm mesh size: a Copenhagen net (in 2010) (*Wiktor, 1982*) and a WP-2 net (in 2011) (HELCOM Manual for Marine Monitoring in the COMBINE Programme of HELCOM, annex C-7).

The plankton net mesh size was selected to collect the mesozooplankton with the younger developmental stages of copepods, which are the main object of the study. A flow meter was placed at 1/4 of the diameter of the net ring in order to determine the amount of water filtered.

The material was collected in accordance with the HELCOM guidelines (Manual for Marine Monitoring in the COMBINE Programme of HELCOM, annex C-7). The vertical net hauls were carried out in the three layers: the bottom—the upper limit of the halocline (with no halocline being present—75 m), the upper limit of the halocline —thermocline (with no thermocline being present—25 m), and the upper limit of the thermocline—the surface. A total number of 21 samples was collected, both during the daytime and night-time.

Table S1 presents a list of samples collected at P1 Station. The division into seasons used in Table S1 and in the following part of the study was adopted based on the water temperature.

The analysed material from the Gdansk Deep was used to determine the composition and seasonal changes in the abundance and biomass in relation to time and space.

The second series of the study material consisted of the monthly zooplankton samples collected in the western part of the Gulf of Gdansk (54°32′N, 18°48.2′E) (Fig. 1, P2 Station) from 11th February, 2010 to 29th November, 2011 from the ship of the Institute of Oceanography of the University of Gdańsk k/h "Oceanograf 2". The location chosen for collection of the plankton samples was characterized by the depth of 40 m and was located approx. 18 km away from the shore. The vertical net hauls were carried out throughout the

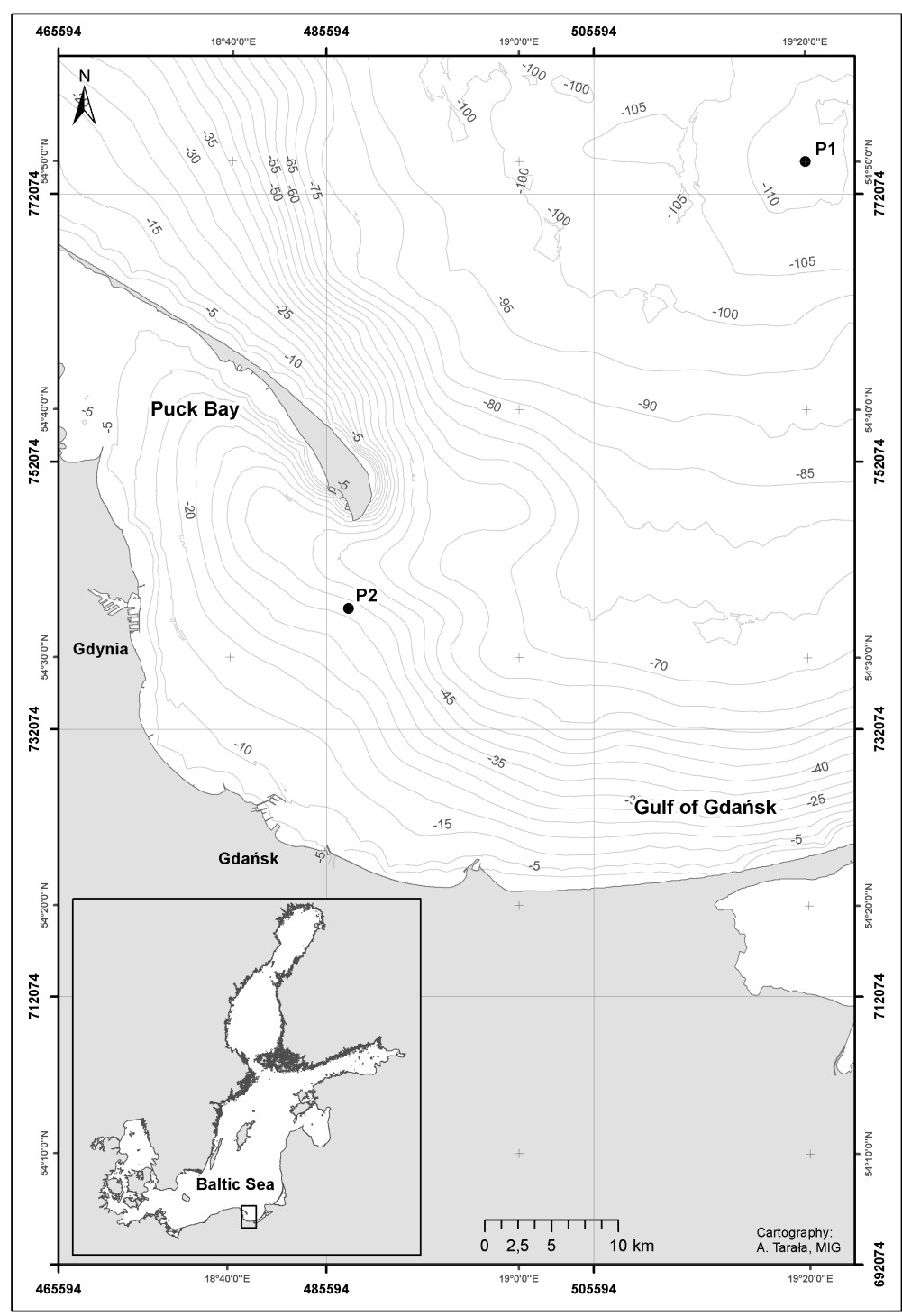

**Figure 1  Map showing the approximate position of the sampling stations: station P1—Gdańsk Deep and P2—inner Gulf of Gdańsk.** Gulf of Gdańsk, southern Baltic Sea in 2010—2011. Credit: Anna Tarała, Maritime Institute in Gdansk.

water column and divided into 10 m thick layers, from the bottom up to the water surface. An exception took place on the 27th of July 2011, when the samples were collected from the following layers: 20–0, 30–20 and 40–30 m, due to an equipment failure. In total, 71 samples were collected in this series.

Table S2 presents a list of samples collected at P2 Station.

The net hauls were carried out only during the day, using (as in the Gdansk Deep in 2011) a WP-2 closing net with 100 µm mesh size. The flow meter was placed at 1/4 of the net ring in order to determine the amount of water filtered. The collected material was immediately transferred into plastic bottles and treated with 4% solution of formaldehyde to preserve animals for subsequent analysis. A total of 92 samples was analysed to the lowest possible taxonomic level, the copepodite stages of different copepod taxa were identified, and copepod nauplii were assigned to the taxa.

The species abundance represents the sum of all development stages in the entire water column. The data for different vertical layers was calculated as the mean value $[m^{-3}]$ from depth-stratum three (P1 Station) or four (P2 Station).

The biomass was calculated from the abundance of weight standards after *Hernroth (1985)*.

The environmental data at the Gdansk Deep (P1), water temperature and salinity were measured in the whole water column using the CTD-probe. Measurements were performed from the r/v ''Oceania'' during seven cruises, prior to the biological material collection.

The environmental data on the western, inner part of the Gulf of Gdansk (P2) came from direct measurements made with a portable meter for analysing water parameters, and were carried out on board the k/h ''Oceanograf-2'' (16 cruises), and on board the ''Hestia'' (two cruises). The measurements were made for each depth-stratum separately.

## RESULTS

### Environmental conditions during the study period

Measurements of the hydrometeorological conditions, taken during the biological material sampling (from January 2010 to November 2011), represent an environmental description within a specific time and space frame (Data S1).

In February 2010 the water temperature at the Gdansk Deep (P1) ranged from 1.8 °C to 9.1 °C on the surface, and the upper limit of the thermocline was determined at a depth of approximately 60 m. While in June (the same year), the measured water temperature ranged from 10.7 °C on the surface to approx. 13 °C in the deepest measured depth (60 m).

The next year, 2011, in March, the water temperature reached the level of 1.4 °C at the surface and remained constant up to the depth of approx. 65 m, where the thermocline began, and where, below that depth the temperature level significantly increased, reaching the value of 6.2 °C at the bottom. In June 2011, the water temperature was measured only to a depth of approx. 50 m. The temperature dropped with the depth increase, from 15.2 °C at the surface to 6.4 °C at a depth of 50 m. November 2011 was characterized by a high surface water temperature, at 11.5 °C, which remained relatively constant up to a depth of approx. 40 m and then rapidly dropped at the greater depths. Due to strong waves

and surface-water cooling, the thermocline was at a depth of approx. 40–50 m. The water temperature at the bottom was 5.2 °C.

The salinity of surface waters at P1 Station ranged from 7.5 to 6.9 PSU and gradually increased along the depth gradient, reaching a maximum value of 12.6–10.8 PSU at the bottom. In June 2010, the salinity was measured only to a depth of 60 m; it ranged from 7.3 PSU at the surface to 6.2 PSU in the deepest layer. Such a large decline in salinity was probably caused by an inflow of flash flood waves into the Gulf of Gdansk after a disastrous spring inundation in the Vistula drainage basin.

The temperature of surface water and salinity measured in 2010 and 2011 at P1 Station (based on the example from June) was significantly different during these two years. The runoff of flood waters in May 2010 disturbed the thermohaline system in the Gdansk Deep, which was reflected in the warmer layer of less saline water.

The water temperature at P2 Station in the western part of the Gulf of Gdansk was slightly higher for 2010 when compared to 2011.

From January to March, the surface-water temperature (approx. 1 °C) was lower than the temperature at the bottom. It was gradually increasing from April and was higher at the surface than at the bottom. However, the differences throughout the entire water column were less than 1 °C. From July to October, the differences in water temperature became more apparent: in 2010 it ranged from 3.4 to 14.1 °C, and in 2011 from 0.2 to 7.0 °C. In November a constant temperature level was observed throughout the water column, at an average of 8.6 °C in 2010 and 7.3 °C in 2011. The warmest month in both 2010 and 2011 was August (19.4 °C and 18 °C), whereas the coldest were January and March (ranging from 1 to 2.1 °C).

The salinity at P2 Station (depth of 40 m) varied to a small extent, both during the year and throughout the water column. The mean values of water salinity in the western part of the Gulf of Gdansk ranged from 6.7 PSU (in July 2010) to 7.4 PSU (in October 2011). The lowest salinity level was recorded in July 2010 (6.4 PSU), which probably resulted from the inflow of the Vistula flood waters.

## Taxonomic composition of zooplankton

According to our environmental studies conducted in 2010/2011, a total of 24 taxa were identified, including 10 Copepoda, four Rotifera, and seven Cladocera, as well as juveniles of unidentified Ctenophora, larvacean *Fritillaria borealis,* and a few specimens of the amphipod *Hyperia galba*. In addition, larvae of the benthic fauna were identified as ones belonging to Polychaeta, Bivalvia, Gastropoda and Cirripedia. They were not identified to the species level, but generally defined as meroplankton. Ichthyoplankton was not analysed in detail either. The research showed that the taxonomic composition of the holoplankton in the Gulf of Gdansk was similar to that observed in this region for many years. The exceptions are two invasive species of Cladocera, which occurred in the summer of 2010 in the shallow part of the Gulf of Gdansk—*Cercopagis pengoi* and *Evadne anonyx*.

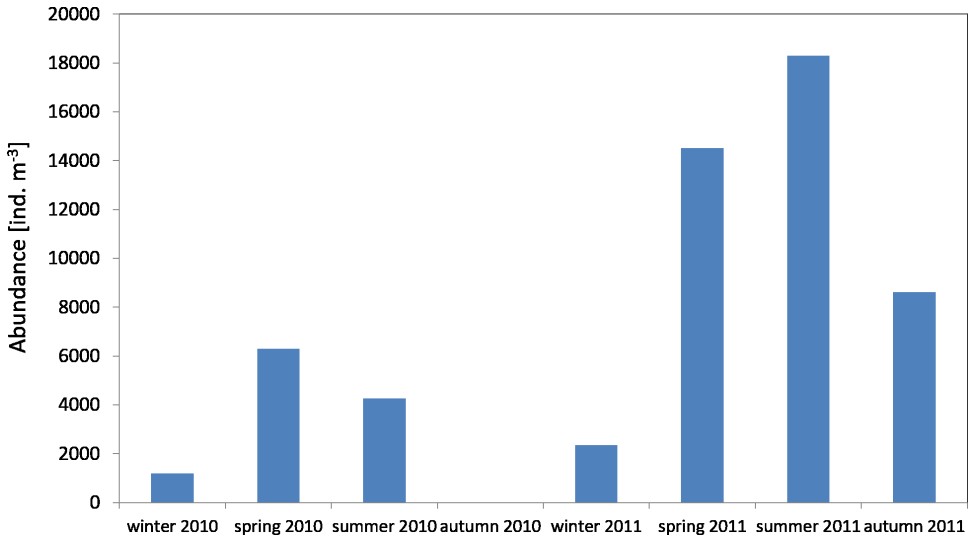

**Figure 2** Abundance of Copepoda; data integrated for the whole water column at station P1 (Gdańsk Deep) in 2010–2011.

## Copepoda abundance
### The open sea waters of the Gulf of Gdansk (Gdansk Deep)—P1

Copepods were usually the main component of zooplankton, in both abundance and biomass (Data S2).

There were limited possibilities for the monthly collection of biological material, and so data collected in select seasons were interpreted as the average of the entire water column, and different developmental stages were summarized for each species. Nevertheless, they provide a general picture of the situation prevailing at a given time in the pelagic zone.

The average number of copepods during the study period of 2010 was 3,913 ind m$^{-3}$ (SD 2,572) and their number ranged from 1,184 ind m$^{-3}$ (in winter) to 6,293 ind m$^{-3}$ (in spring). One year later, the average count of copepods was higher at 11,723 ind m$^{-3}$ (SD 6,980), and it ranged from 2,351 ind mnd m (in winter) to 18,307 ind m$^{-3}$ (in summer) (Fig. 2) (Data S2).

The maximum number of copepods in spring 2010 in the surface layer (25–0 m) was 12,545 ind m$^{-3}$, while in spring 2011 the count of copepods in the same layer was 2.5 times higher. In the other months, the highest values of the copepods count were also recorded in the layer between the upper limit of the thermocline and the surface (Data S2).

The *Pseudocalanus* sp. had a higher relative proportion at P1 Station than at P2 Station (Figs. 3–5). This species was the main component of copepods in the winter-spring season of 2010 (approx. 50%), while it was replaced by *Acartia* spp. (40.26%) and *T. longicornis* (33.31%) in summer. During this period, *C. hamatus* accounted for several percent of the copepods, while *Eurytemora* sp. was insignificant.

In 2011, the situation was similar, as *Pseudocalanus* sp. was the main component of copepods abundance in the winter-spring season (over 50%), while in the summer-autumn season its contribution dropped and was similar to that of *T. longicornis*—40% in summer

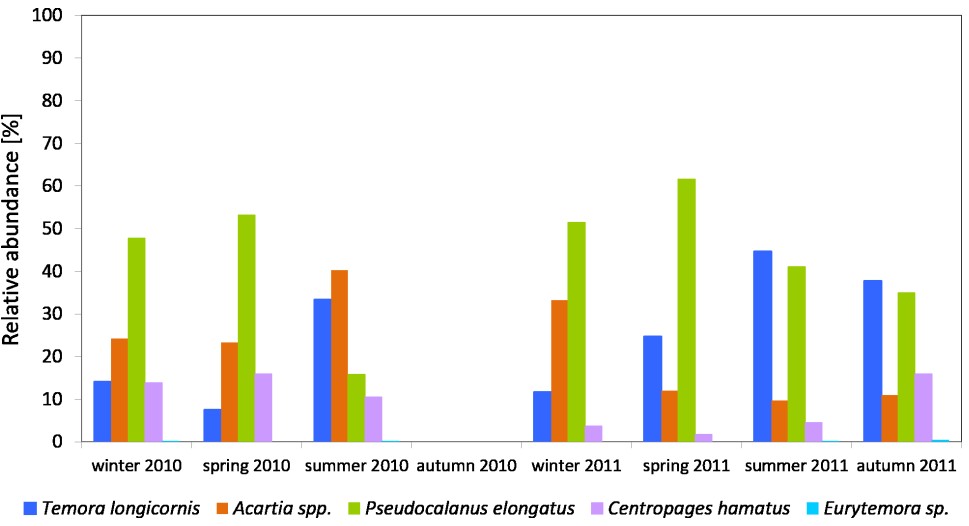

**Figure 3** **Taxonomic structure of Copepoda abundance at station P1 (Gdańsk Deep; data integrated for the whole water column) in 2010–2011.**

and 35% in autumn. The percentage of the genus *Acartia* in the described seasons ranged from 10 to 30%. The presence of *C. hamatus* in this region ranged from a few to several percent, while the count of *Eurytemora* sp. (similarly to the previous year) was insignificant (Fig. 3).

### The coastal, inner waters of the Gulf of Gdansk—P2

Copepods usually represented the main component of zooplankton. Their average count in 2010 was 29 141 ind m$^{-3}$ (SD 23315), and ranged from 3,330 ind m$^{-3}$ (in March) to 67,789 ind m$^{-3}$ (in May). The average count of copepods in 2011 was much lower at 17,883 ind m$^{-3}$ (SD 11,407), and ranged from 1,360 ind m$^{-3}$ (in April) to 39,558 ind m$^{-3}$ (in May) (Fig. 4) (Data S3).

The maximum count of copepods in May 2010 was determined in the 10-0 m layer, at 161,150 ind m$^{-3}$, while in September 2011 the copepods abundance in the same layer was less than half of that (70,314 ind m$^{-3}$) Data S5.

The analysis of seasonal changes in 2010 revealed two peaks in Copepoda abundance: the first in May and the second in September, with an abundance of 67,789 ind m$^{-3}$ and 57,822 ind  m$^{-3}$, respectively. In 2011, there was a large peak of abundance in September (39,559 ind m$^{-3}$) (Fig. 6).

It appears that the distribution of copepods in the water column is determined by the preferences of a species dominant at a given time and its developmental stage.

In March and June 2010, the largest number of copepods was observed in the 30–20 m layer, while in April this was observed in the 20–10 m layer. During the rest of the year, copepods occurred mainly in the surface layer (10–0 m) (Fig. S1 ). In 2011, the situation was slightly different. In the early spring and autumn, the largest numbers of copepods

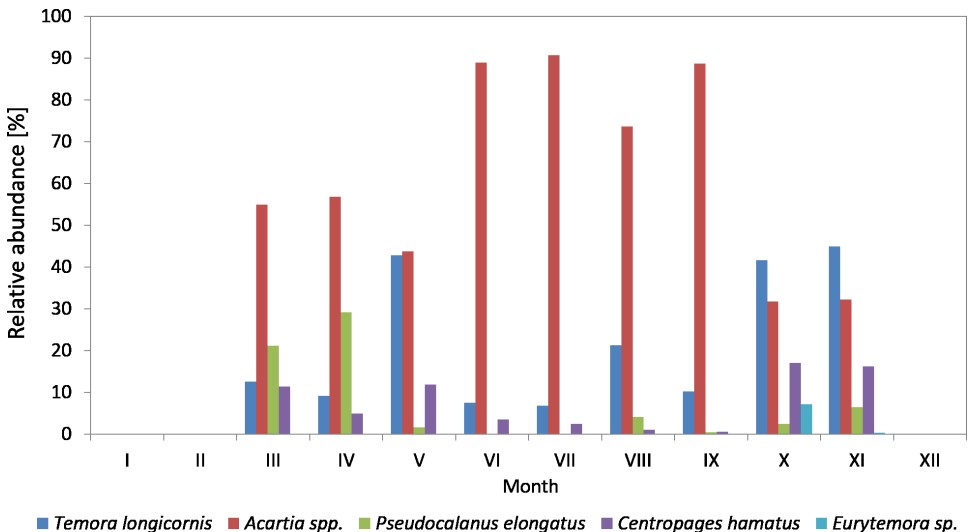

**Figure 4** **Taxonomic structure of Copepoda abundance at station P2 (inner Gulf of Gdańsk; data integrated for the whole water column) in 2010.**

were observed in layers up to 20 m, whereas in summer, they definitely preferred deeper waters (Fig. S2) (Data S5).

In 2010, the genus *Acartia* was the main component of copepods in the period from March to September, ranging from 26.23 to 89.38%, while in October and November it was at the level of 32% (Fig. 4) (Data S3).

*T. longicornis* was the second most abundant Copepoda species—from 6.85% (in July) to 44.90% (in November). In October and November, *T. longicornis* dominated and in May its abundance was only slightly lower compared to *Acartia* spp.

The contribution of *Pseudocalanus* sp. was also relatively significant and ranged from 21.16% in March to 29.16% in April. During the rest of the year, it ranged from only 0.07 (in June) to 6.43% (in November).

The abundance of *C. hamatus*, similarly to *Pseudocalanus* sp., was higher in the spring and in the autumn and ranged from 11.38% in March to 16–17% in October and November.

On the other hand, the contribution of *Eurytemora* sp. did not exceed 1% throughout the study period, except for October 2010 when it reached approx. 7%.

In 2011, the genus *Acartia* accounted for the largest contribution in the abundance of copepods (except for May and June), ranging from 15.81% (in May) to 85.25% (in August) (Fig. 5) (Data S3).

*T. longicornis* was the dominant species among copepods in May (77.19%) and June (58.42%). Its contribution in July, October and November was approximately 30%, and in the other months just several percent.

Similarly to the previous year, *Pseudocalanus* sp. was the most abundant Copepoda species in April (23.45%), while in the other months it accounted for up to 7%.
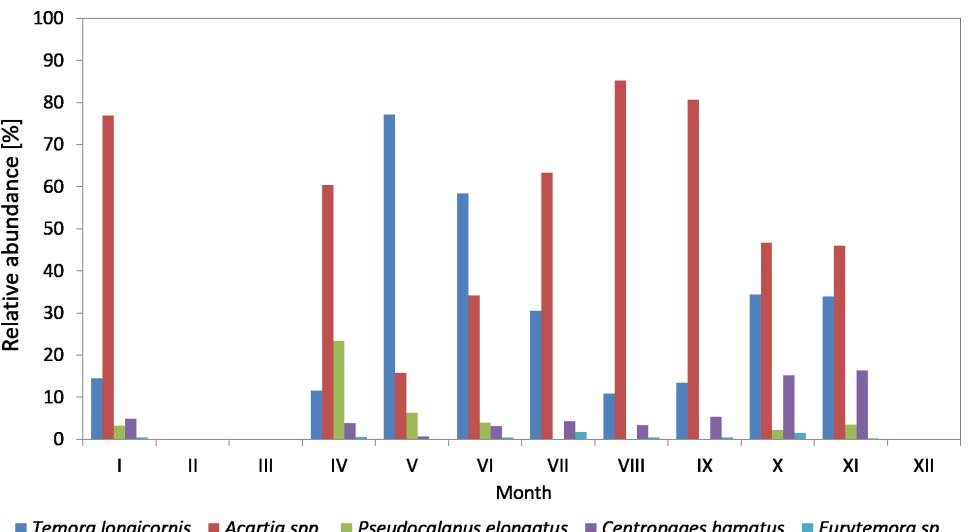

**Figure 5** Taxonomic structure of Copepoda abundance at station P2 (inner Gulf of Gdansk; data integrated for the whole water column) in 2011.

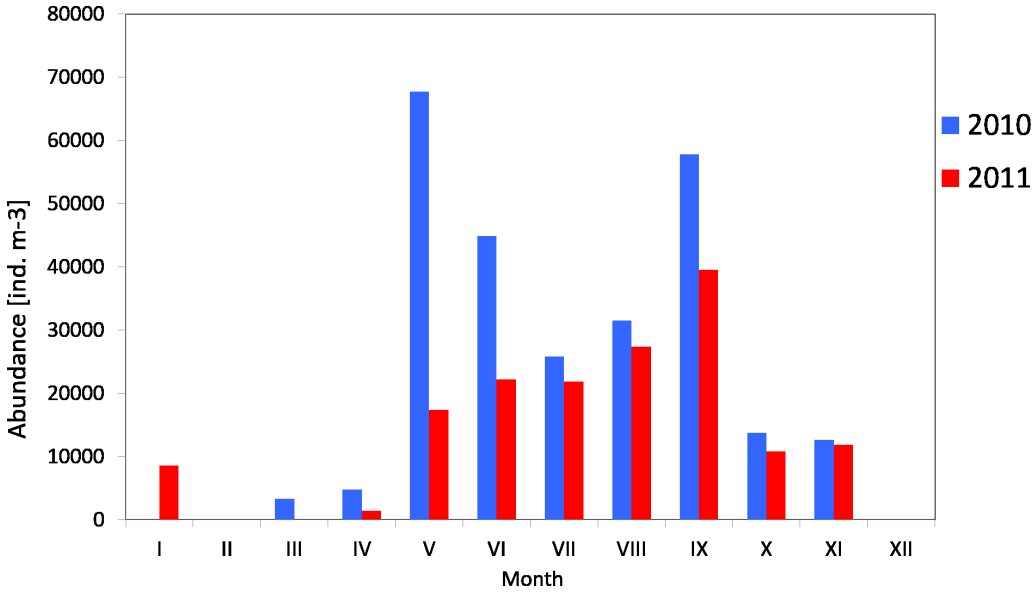

**Figure 6** Abundance of Copepoda; data integrated for the whole water column at station P2 (inner Gulf of Gdańsk) in 2010–2011.

*C. hamatus* was a significant component of copepods in October (15.16%) and November (16.33%), the same as in the autumn of 2010. Its contribution was insignificant for most of the year, ranging from 0.65 (in May) to 5.34% (in September).

In 2011, *Eurytemora* sp. was rare, only accounting for up to 1.63% (July) of the total count of copepods.

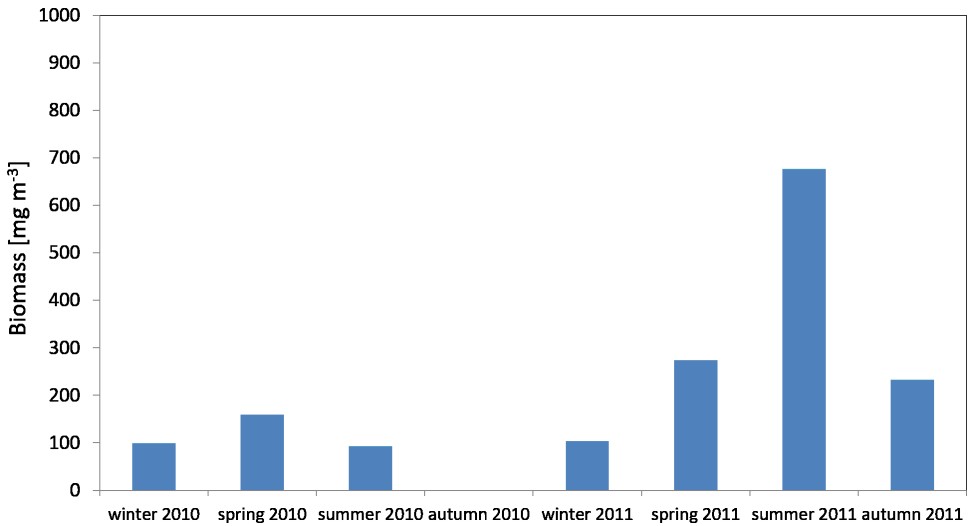

**Figure 7** Biomass of Copepoda; data integrated for the whole water column at station P1 (Gdańsk Deep) in 2010–2011.

## Copepoda biomass
### The open sea waters of the Gulf of Gdansk (Gdansk Deep)—P1
The average biomass of copepods in the zooplankton in 2010 at P1 Station was about 116.68 mg m$^{-3}$ (SD 37.49) and ranged from 92.19 mg m$^{-3}$ (in summer) to 159.84 mg m$^{-3}$ (in spring), while in 2011 the average value was 321.26 mg m$^{-3}$ (SD 247.418), and ranged from 103.67 mg m$^{-3}$ (in winter) to 676.20 mg m$^{-3}$ (in summer) (Fig. 7) (Data S4).

The maximum biomass of copepods in spring 2010 was recorded in the surface layer (up to a depth of 25 m), at 83.59 mg m$^{-3}$, and in summer 2011 in the intermediate layer (from the upper limit of the halocline to the upper limit of the thermocline, i.e., 70–25 m), at 467.07 mg m$^{-3}$ (Data S4).

Considering the contribution of individual Copepoda taxa in the zooplankton biomass at P1 Station, one can observe a clear dominance of *Psedocalanus* sp., which accounted for about 50% of the total biomass of copepods in the winter–spring season of 2010, while in summer 2010 its abundance dropped in favour of *T. longicornis* and *Acartia* spp. (approx. 40%). The abundance of *C. hamatus* also increased in the spring season up to 23.64%.

In the winter–spring season of 2011, *Psedocalanus* sp. represented approximately 60% of the Copepoda biomass, while in summer its contribution dropped to 22.88% and again increased to 47.97% in autumn. *T. longicornis* (65.05%) was the main component of the Copepoda biomass in summer. The contribution of other species was negligible: *Acartia* spp. from 8.80 to 13.33% and *C. hamatus* from 3.22 to 10.24% (Fig. 8) (Data S4).

### The coastal, inner waters of the Gulf of Gdansk—P2
The average biomass of copepods at station P2 in 2010 was 151.46 mg m$^{-3}$ (SD 115) and it ranged from 33.87 mg m$^{-3}$ (in March) to 390.12 mg m$^{-3}$ (in May). In 2011, the average

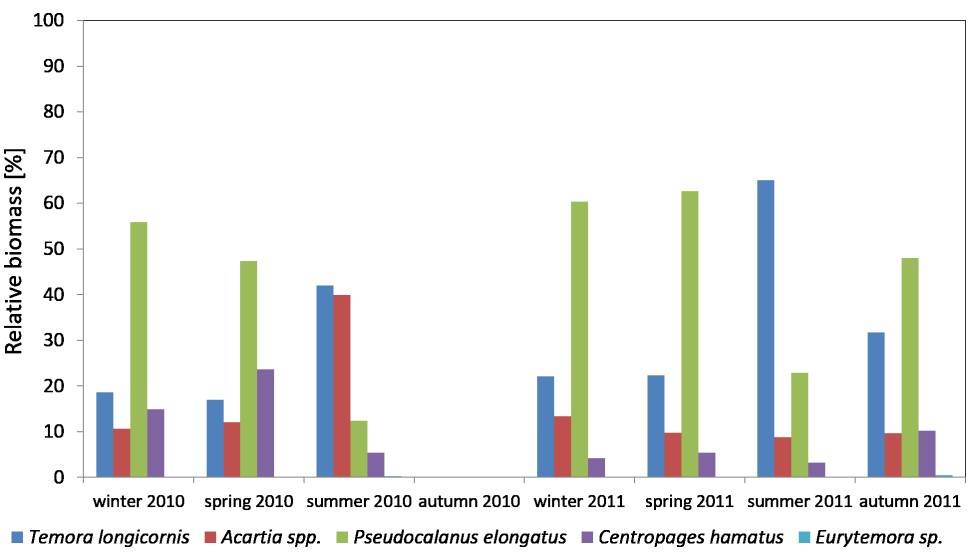

**Figure 8** Taxonomic structure of Copepoda biomass at station P1 (Gdańsk Deep; data integrated for the whole water column) in 2010–2011.

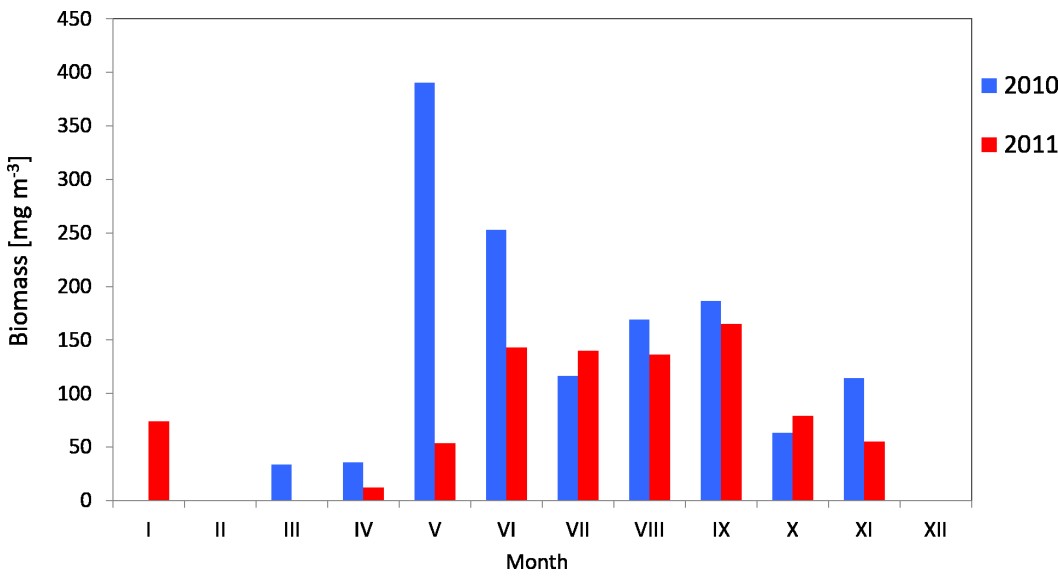

**Figure 9** Biomass of Copepoda; data integrated for the whole water column at station P2 (inner Gulf of Gdańsk) in 2010–2011.

Copepoda biomass was 95.47 mg m$^{-3}$ (SD 52) and ranged from 12.40 mg m$^{-3}$ (in April) to 164.82 mg m$^{-3}$ (in September) (Fig. 9) (Data S3).

The maximum biomass of copepods in May 2010 was recorded in the 10–0 m layer, at 692.12 mg m$^{-3}$, and at 403.98 mg m$^{-3}$ in September 2011.

When looking into the seasonal changes in the Copepoda biomass in 2010, it appears that a significant peak occurred in May, at 390.12 mg m$^{-3}$, and that two smaller peaks

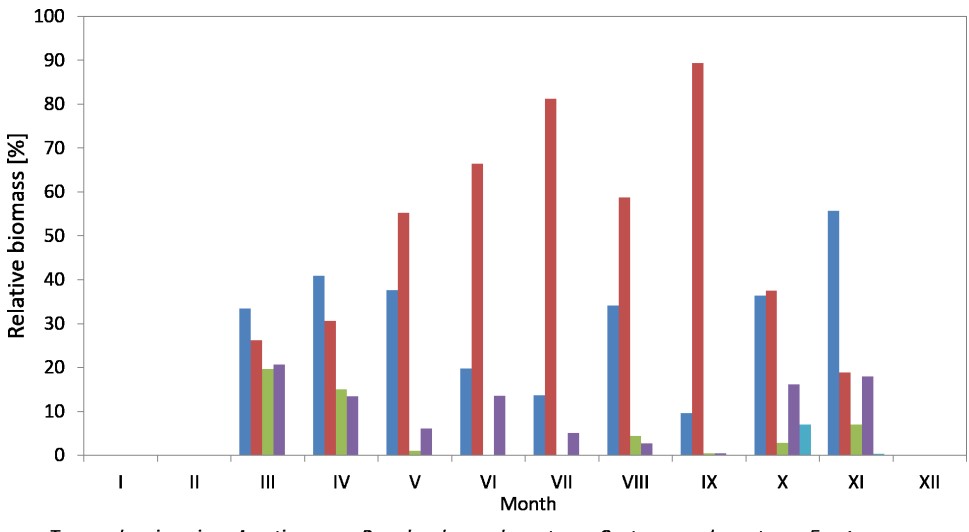

**Figure 10** Taxonomic structure of Copepoda biomass at station P2 (inner Gulf of Gdańsk; data integrated for the whole water column) in 2010.

occurred both in September (186.73 mg m$^{-3}$) and November (114.36 mg m$^{-3}$). In 2011, biomass levelled from June to September, reaching its maximum in September (164.82 mg m$^{-3}$) (Fig. 9).

In March and June 2010, the largest number of copepods was observed in the 30–20 m layer, while in April 2010 it was in the 20–10 m layer. In the other months, the highest values of biomass were determined in the surface layer (10–0 m) (Fig. S3) (Data S5).

In 2011, the biomass values had a similar pattern, except for January and October when the values were slightly higher at the bottom (40–30 m) (Fig. S4) (Data S5).

The species from the genus *Acartia* spp. dominated in the biomass of copepods at P2 Station for the most of the 2010 season. Their contribution ranged from 18.92% in November to 89.38% in September. In March, April and November, they were replaced by *T. longicornis*. In October, the biomass of both taxa was at a similar level, approx. 37%.

*T. longicornis* was a subdominant in the biomass of copepods. Its contribution ranged from 9.65% (in September) to 55.69% (in November).

As in the case of abundance, a significant percentage of *Pseudocalanus* sp. in the biomass of copepods was observed only in March (19.63%) and April (15%), while in the remaining months it ranged from only 0.10% (July) to 7.04% (November).

*C. hamatus* was a constant component of the Copepoda biomass, with the highest values recorded in March (20.74%), April (13.49%), June (13.55%), October (16.12%) and November (17.98%). The percentage of *Eurytemora* sp. in the Copepoda biomass was usually up to 1%, except for October at 7.05% (Fig. 10) (Data S3).

In 2011, the genus *Acartia* represented a significant component of the copepods biomass with the contribution ranging from 12.87% (in May) to 88.16% (in August).

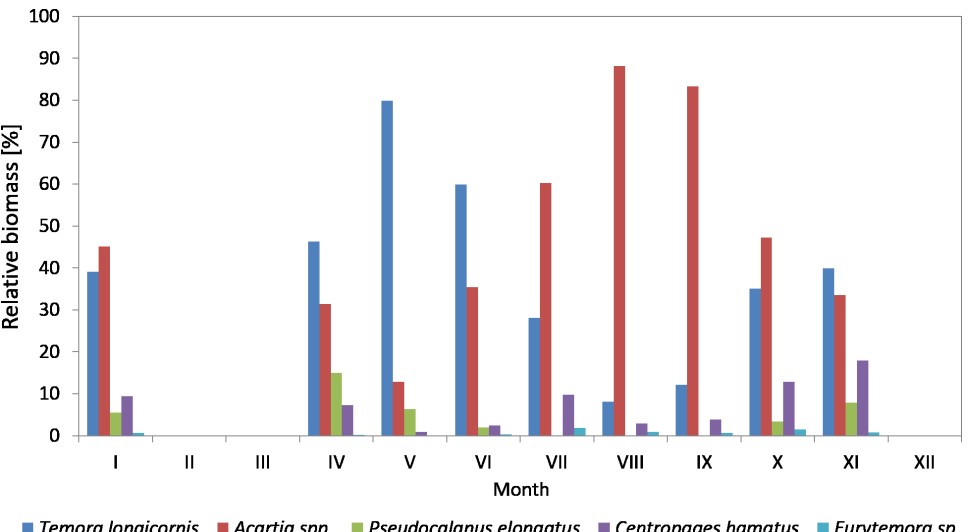

**Figure 11  Taxonomic structure of Copepoda biomass at station P2 (inner Gulf of Gdańsk; data integrated for the whole water column) in 2011.**

*T. longicornis*, being an important and constant component in the biomass of copepods, was observed from April to June, and then in November, ranging from 8.05% (in August) to 79.87% (in May). In January and November, the biomass values of *Acartia* spp. and *T. longicornis* were similar.

The maximum biomass of *Pseudocalanus* sp. was determined in April, at 14.94%, while for the rest of the year the biomass values were low. In 2011, the crustacean *C. hamatus* was much more important in the biomass of copepods, at 10% in January, July, October and November, and ranging from 0.89% in May to 7.23% in April. *Eurytemora* sp. were of minor significance in the biomass of Copepoda, the same as in the previous year (Fig. 11) (Data S3).

## DISCUSSION

In the terms of biomass and abundance, Copepoda are the most important zooplankton taxa in the southern Baltic, and they are mainly represented by e.g., *Acartia* spp., *Pseudocalanus* sp. and *T. longicornis*. Rotifera: *Synchaeta* spp., *Keratella quadrata*, and Cladocera: *Evadne nordmanni*, *Bosmina (Eubosmina) coregoni* and *Pleopis polyphemoides*, are more important in the coastal regions, while euryhaline freshwater and typically freshwater species (e.g., *Eurytemora* sp.) occur mainly at the river mouths and are of lesser importance.

Copepods represent one of the largest groups of secondary producers in the World Ocean. They are an important link between phytoplankton, microzooplankton and higher trophic levels such as fish (*Longhurst, 1981*; *Longhurst & Harrison, 1989*; *Kleppel, Holliday & Pieper, 1991*; *Kleppel, 1992*; *Dzierzbicka-Glowacka et al., 2011*). They are an important source of food for many fish species, but also a significant producer of detritus. One individual organism can produce 200 portions of fecal matter per day, which is an

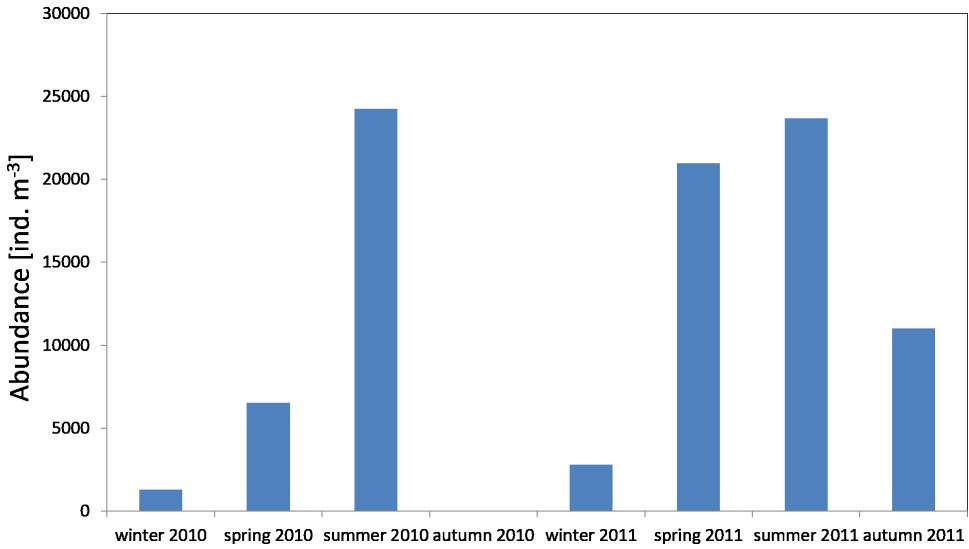

**Figure 12** **Abundance of zooplankton; data integrated for the whole water column at station P1 (inner Gulf of Gdańsk) in 2010–2011.**

important source of food for detritivores and is very important in both sedimentation and circulation of biogenic substances processes.

The study presents an analysis of 92 zooplankton samples from the Gdansk Deep (Gdansk Basin) and from the western part of the Gulf of Gdansk in the terms of composition, abundance and biomass of zooplankton, with particular emphasis on copepods, as well as on the structure of populations of species occurring in large numbers in the southern Baltic, i.e., *Pseudocalanus* sp., *Acartia* spp. *Temora longicorni,* in 2010 and 2011.

## Changes in the abundance and biomass of zooplankton

Taxa occurring in the samples occasionally or in small numbers (*Hyperia galba*, *Oithona similis*, Ctenophora, freshwater Cyclopoida, Harpacticoida) were not included in the determination of zooplankton abundance and biomass.

The taxonomic structure of zooplankton observed during the research period was quite typical for the southern Baltic (*Mudrak & Zmijewska, 2007*). The exception was an invasive species of Cladocera, *Evadne anonyx*, which occurred in the summer of 2010 in the inner Gulf of Gdansk. This was the first, and so far only observation of this species in the Gulf of Gdansk (*Bielecka, Mudrak-Cegiołka & Kalarus, 2014*).

The average count of zooplankton in the Gdansk Deep (P1 Station) during the conducted studies was 10,685 ind per $m^{-3}$ (SD 12,027), whereas in 2011 it was 14,607 ind per. $m^{-3}$ (SD 9565). The highest mean values of abundance in the water column were recorded in the summer seasons of 2010 and 2011, at 24,238 ind $m^{-3}$ and 23,659 ind $m^{-3}$, respectively. The minimum values were observed in the winter–spring season (1,283 ind $m^{-3}$ and 2,807 ind $m^{-3}$) (Fig. 12) (Data S2).

The average count of zooplankton in the western part of the Gulf of Gdansk (at P2 Station) in 2010 was 87,122 ind $m^{-3}$ (SD 104,836), and in 2011 it was 31,649 ind $m^{-3}$

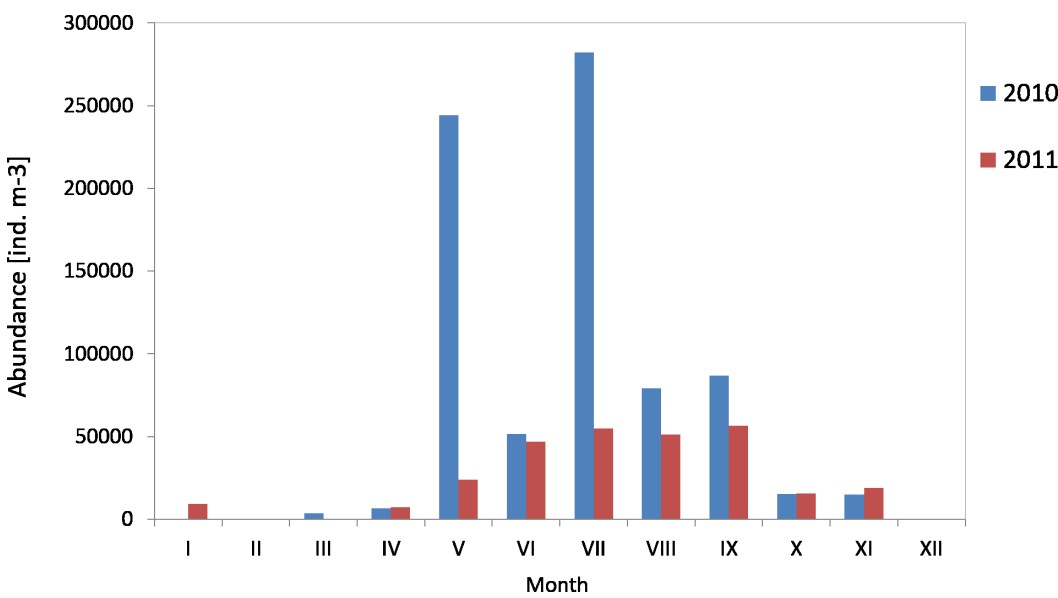

**Figure 13** **Abundance of zooplankton; data integrated for the whole water column at station P2 (Gdańsk Deep) in 2010–2011.**

(SD 20,487). In 2010, the maximum average count of zooplankton in the water column was recorded in July, whereas in 2011 in September it was 282,166 ind m$^{-3}$ and 56,657 ind m$^{-3}$, respectively. The minimum values were recorded in March 2010 (3,617 ind m$^{-3}$) and April 2011 (7,249 ind m$^{-3}$) (Fig. 13) (Figure S1).

The zooplankton at P1 Station varied, depending on the seasons, although not as much as in the shallow regions of the Gulf of Gdansk. In the two-year cycle of the scientific studies, copepods were the main component of zooplankton, representing from 69% of the total zooplankton in spring 2011 to 96% in spring 2010 (except for the summer in 2010, approx. 18%) Data S2.

In 2010 and 2011, copepods occurred at P2 Station throughout the study period and for most of the months they were the main component of zooplankton, with the contribution ranging from approx. 67% (in September) to 92% (in March) in 2010 and from 47% (in June) to 93% (in January) in 2011, except for May (24%) and July (over 9%), when rotifers dominated in the zooplankton. In August, the contribution of Copepoda was similar to Cladocera and Rotifera and amounted to approx. 40%. In 2011, the exceptions were April and July, when pelagic fauna was dominated by meroplankton, mainly veligers of bivalves Data S3.

Copepods were the main component of the zooplankton biomass at P1 Station for the whole study duration, with the contribution ranging from 55.3% in summer 2010 to 99.2% in winter 2010.

The situation was different at P2 Station. In March, April and June, as well as in September, October and November 2010, Copepoda accounted for the main part of the zooplankton biomass, from 67.6% in October to approx. 94.6% in March. In May, July

and August, as a result of the seasonal zooplankton components occurring during these months (e.g., Cladocera), the proportion of copepods significantly decreased and ranged from 24.2 to 36.7%. In 2011, copepods dominated at P2 Station, and their contribution in the total biomass ranged from 31.7% (in April) to 96.7% (in January). In April, juvenile stages of the benthic fauna dominated in the zooplankton biomass at 64.35%, while in the following months their contribution dropped to 7.03%, and then increased again in July, to 34.95% Data S3.

## Changes in the abundance and biomass of Copepoda

The coastal region of the Gulf of Gdansk is wide open towards the Gdansk Deep, which is part of the Gdansk Basin, the southernmost part of the Gotland Basin, which is the largest and the deepest basin of the Baltic Sea.

The vertical profile of waters in the Gulf of Gdansk can be divided into two layers. The surface layer in the coastal area reaches the bottom. In the deeper part, it is separated from the lower layer by the intermediate waters, up to 60–80 m in depth. The surface layer is subject to seasonal changes in temperature, caused by meteorological factors, convection, wind mixing and the impact of the Vistula River water, which causes warming in spring and summer and cooling in autumn and winter seasons. The impact of the Vistula River varies during the year: in spring and summer seasons it covers almost the entire gulf while in November it is limited to estuaries. This is due to the force and the direction of winds. There is a difference in the vertical distribution between the coastal and the deep-sea regions. The coastal areas have higher temperatures in summer compared to the surrounding waters, while in winter they are cooler. The annual report shows that the salinity in the Gulf of Gdansk is lower in winter than in summer. A key factor affecting the salinity of the surface waters are fresh waters from the Vistula River.

The environmental conditions of the pelagic habitat change with both depth and distance from the shore. Although the qualitative (taxonomic) structure of zooplankton is almost identical with that of the coastal waters, the quantitative structure (abundance and biomass) changes quite significantly. The maximum values of zooplankton abundance and biomass were observed in the summer season, both in the Gdansk Deep and in the inner part of the Gulf of Gdansk. Copepods dominated in the composition of zooplankton for almost the entire duration of the research. Quantitative composition of copepods at P1 Station differed from that at P2 Station due to the high abundance of *Pseudocalanus* sp., which prefers colder, more saline waters. Other species typical in colder and more saline waters were *Acartia longiremis,* which was also more abundant at lower water levels, while *A. bifilosa* and *A. tonsa* were more common in the surface waters and in the coastal area. In general, *Acartia* species distribution is strongly connected with salinity, and with decreasing salinity from the open part of the Baltic Sea towards both the Gulf of Finland and the Gulf of Bothnia, the proportion of *Acartia* decreases while *Eurytemora* increases (*Simm & Ojaveer, 2000*). In the gulf of Riga *Acartia bifilosa* dominates in winter-spring (it makes up 58–68% of total zooplankton) while *Eurytemora* becomes the dominant species during summer (*Ojaveer, 2017*). In the Gulf of Gdansk *Eurytemora* plays only a minor role and the contribution to the total zooplankton biomass is less than 10% over the whole year.

In the open part of the Gulf of Gdansk the concentration of *Pseudocalanus* sp. and *T. longicornis* was higher compared to the coastal station (*Dzierzbicka-Glowacka, Kalarus & Zmijewska, 2013*; *Dzierzbicka-Glowacka et al., 2015*). In agreement with another study (*Ackefors, 1981*) *Pseudocalanus* tend to dominate in offshore areas of the Baltic Sea, rather than in the coastal zones, and it is rarely present above the thermocline during the warm summer period. In the offshore deep areas, this genus can constitute 70–100% of all copepods in the water column (*Ackefors & Hernroth, 1972*).

In both 2010 and 2011, *Pseudocalanus* sp. was the main component of copepod biomass and abundance at the P1 Station in the winter-spring season (approx. 50% and 60% of the abundance and biomass, respectively), and in the summer-autumn season its contribution dropped and was similar to that of *T. longicornis*: approx. 40% in summer and 35% in autumn. The percentage of the genus *Acartia* in these seasons ranged from 10% to 30%.

The analysis of the variation in the Copepoda taxonomic structure in the inner part of the Gulf of Gdansk at P2 Station indicates that *Acartia* spp. dominated in the copepods composition. Its contribution in 2010 ranged from 26% (in March) to 89% (in September), and in 2011 it ranged from 16% (in May) to 85% (in August), while in October and November it was approx. 32%.

*T. longicornis* was a sub-dominant species in the terms of abundance and biomass of copepods in the Gulf of Gdansk. Its maximum contribution in the total abundance at P2 Station was approx. 45% (in November 2010), 77% (in May 2011) and approx. 56% (in November 2010) and 80% (in May 2011) in the biomass.

## Abundance, comparison with other data

Taking into account the two periods—2006/2007 (*Dzierzbicka-Glowacka, Kalarus & Zmijewska, 2013*) and 2010/2011—the total count of copepods in the Gdansk Basin (at P2 Station) was characterized by a significant increase (three- and twofold) in the maximum abundance within the 10–0 m layer in May 2010 (161,150 ind m$^{-3}$) and within the 20–10 m layer in July 2007 (127,000 ind m$^{-3}$), as well as in the average value in the water column, at 67,790 ind in May 2010 and 83,500 ind in July 2007. In 2014, in the Lithuanian Baltic Sea, at the coastal stations (B1–B4) and at the open sea stations (B5–B9) (Table 1), the average abundance of copepods in the surface layer (36,320 and 21,327 ind m$^{-3}$) was similar to the values from 2011 and 2006 for the Gulf of Gdansk and they were about two and four times lower than in 2011 and 2010, respectively (Table 1).

In general, the maximum contribution (%) of *Acartia* spp., *T. longicornis* and *Pseudocalanus* sp. in the abundance of Copepoda at P2 Station in the western part of the Gdansk Gulf was similar in the two periods of 2006/2007 and 2010/2011 (Table 2). The population dynamics of the main Baltic calanoid copepod species in the Gdansk Basin in the two study periods was characterized by an increase in the maximum percentage contribution of *Acartia* spp. (up to 90%) and *Pseudocalanus* sp. (up to 29%) and by a decline of *T. longicornis* (to 45%) in the abundance of copepods in 2010, and a major growth (up to 77%) of *T. longicornis* in 2011, as well as a decline of *Acartia* spp. and *Pseudocalanus* sp. in 2011, to the level from the period of 2006/2007. In the other cases, the percentage of individual taxa was at a similar level at all of them, throughout the study period.

**Table 1** Abundance (ind m$^{-3}$) max (left column) and mean (right column) of *Acartia* spp., *T. longicornis*, *Pseudocalanus sp.* at station P2 in the inner Gulf of Gdańsk and at stations B1–B4 and B5–B9 in Lithuanian waters. Data from the inner Gulf of Gdańsk for 2006 and 2007 (*Dzierzbicka-Glowacka, Kalarus & Zmijewska, 2013*), 2010 and 2011 (Data S2–S5). Data from Lithuanian waters coastal stations (B1–B4)* and open sea stations (B5–B9)** for 2014 (E Griniené, 2014, unpublished data).

| Year | Max ind m$^{-3}$ (month and layer) | Average ind m$^{-3}$ (month) |
|---|---|---|
| 2006 | 57,500 (July in 40–30 m ) | 25,600 (June) |
| 2007 | 127,000 (July in 20–10 m) | 83,500 (July) |
| 2010 | 161,150 (May in 10–0 m) | 67,790 (May) |
| 2011 | 70,300 (Sept. in 10–0 m) | 39,560 (Sept.) |
| 2014 * | 40,317 (July in 25–0 m) | 36,320 (July) |
| 2014 ** | 43,912 (July in 25–0 m) | 21,327 (July) |

**Table 2** Maximum contribution (in %) of *Acartia* spp., *T. longicornis* and *Pseudocalanus* sp. to the total abundance of Copepoda at stations P1 and P2 in the Gulf of Gdańsk and at stations B1–B4 and B5–B9 in Lithuanian waters. Data from the inner Gulf of Gdańsk for 2006 and 2007 (*Dzierzbicka-Glowacka, Kalarus & Zmijewska, 2013*), 2010 and 2011 (Data S2–Data S5). Data from Lithuanian waters coastal stations (B1–B4)* and open sea stations (B5–B9)** for 2014 (E Griniené, 2014, unpublished data). [ ] - Max % in separate station from stations B1–B4 and B5–B9; ( ) - averaged per stations B1–B4 and B5–B9.

| | *Acartia* spp. | *Temora longicornis* | *Pseudocalanus* sp. |
|---|---|---|---|
| 2006 (P2) | 86 (Sept.) | 57 (Nov.) | 25 (Feb.) |
| 2007 (P2) | 82 (Aug. and Sept.) | 51 (June) | 25 (March) |
| 2010 (P2) | 90 (June, July and Sept.) | 45 (Nov.) | 29 (April) |
| 2011 (P2) | 85 (Aug) | 77 (May) | 23 (April) |
| 2010 (P1) | 40 (June) | 33 (June) | 53 (April) |
| 2011 (P1) | 33 (March) | 45 (June) | 62 (May) |
| 2014* | [57](43) (July) | [66] (47) (July) | [39] (13) (April) |
| 2014** | [59] (39) (April) | [71] (56) (July) | [69] (37) (April) |

The taxon *Acartia* spp. had the highest percentage contribution (approx. 82–90%) in all the studied years, particularly in the summer season (June–September). *T. longicornis* accounted for approx. 45–57% (i.e., almost half the *Acartia* abundance) of the total Copepoda abundance in the studied periods (2006/2007 and 2010/2011; except for 2011—with 77%), i.e., late spring/summer season (May/June) and autumn season (November) when, soon after or before these periods, *Acartia* spp. reached the first of its second peak in abundance, respectively. On the other hand, the highest contribution of *Pseudocalanus* sp., the third most abundant copepod species in the inner part of the Gulf of Gdansk, was observed in the early spring (March/April at approx. 23–29%), except for in 2006 (in February at 25%). *Pseudocalanus* sp. is a typical representative of the winter zooplankton. Outside the winter season, the taxon is present mostly in cooler, deep-water layers in the Gulf of Gdansk (*Siudziński, 1977*).

At P1 Station (The Gdansk Deep) in 2010 and 2011, the maximum contribution (%) of *Acartia* spp. (40–33%) was similar to that of *T. longicornis* (33–45%) and two times lower

than at P2 Station. However, *Pseudocalanus* sp., had the highest percentage contribution (approx. 53–62%), particularly in the spring time (April).

The percentage contribution which has been observed for this species in the Gulf of Gdansk (P1) was similar to that which has been observed in the Lithuanian Baltic Sea on the open sea stations (B5–B9): for *Acartia* spp. and *T. longicornis* for the average values (in ( )), in turn for *Pseudocalanus* sp. for the maximum value (in [ ]) (Table 2).

## CONCLUSION

The taxonomic composition of zooplankton in the Gulf of Gdansk appears to be stable. An additional difficulty in comparing the data from different years results mainly from the various sampling methods, especially the mesh size of the sampling nets used. It appears that the high similarity of zooplankton composition between the Gdansk Deep and the more coastal part of the inner Gdansk Gulf confirms that the latter is highly influenced by the open sea waters of the Baltic proper. This makes it different from the other large Baltic gulfs, the Gulfs of Bothnia, Finland and Riga, which differ from the Baltic proper by their biotic and abiotic characteristics and are often categorized as their own autonomous subunits (*Ojaveer & Elken, 1997*).

Thorough knowledge of the species composition, the dominance of particular taxa, density and biomass—in combination with abiotic—makes it easier to assess changes which take place in the ecosystem. In combination with the simulation models, such knowledge provides hypothetical forecasts for the future, leading to anticipation of positive or negative effects of environmental changes.

## ACKNOWLEDGEMENTS

We express our gratefulness to the anonymous reviewers for their valuable comments on the earlier versions of the manuscript.

### Funding

This work was supported by the European Union through European Regional Development Fund within Pomorskie Voivodeship Regional Operational Programme for 2014-2020 (FindFISH No. RPPM.01.01.01-22-0025/16-00) and the National Centre for Research and Development within the BIOSTRATEG III program No. BIOSTRATEG3/343927/3/NCBR/2017. The funders had no role in study design, data collection and analysis, decision to publish, or preparation of the manuscript.

### Grant Disclosures

The following grant information was disclosed by the authors:
European Union through European Regional Development Fund: RPPM.01.01.01-22-0025/16-00.
National Centre for Research and Development: BIOSTRATEG3/343927/3/NCBR/2017.

## Competing Interests

The authors declare there are no competing interests.

## Author Contributions

- Lidia Dzierzbicka-Glowacka conceived and designed the experiments, performed the experiments, analysed the data, prepared figures and/or tables, authored or reviewed drafts of the paper, approved the final draft.
- Anna Lemieszek conceived and designed the experiments, performed the experiments, analysed the data, contributed reagents/materials/analysis tools, prepared figures and/or tables, authored or reviewed drafts of the paper, approved the final draft.
- Marcin Kalarus performed the experiments, analysed the data, prepared figures and/or tables, approved the final draft.
- Evelina Griniene performed the experiments, prepared figures and/or tables, approved the final draft.

## Data Availability

The raw data are provided in Supplemental File.

## Supplemental Information

Supplemental information for this article can be found online at http://dx.doi.org/10.7717/peerj.5562#supplemental-information.

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
