# Peer review of "Seasonal changes in the abundance and biomass of copepods in the south-eastern Baltic Sea in 2010 and 2011"

_PeerJ, doi:10.7717/peerj.5562_

## Round 0.1 · original submission · Major Revisions

Both reviewers found the data presented valuable, but consider the manuscript needs to be improved in several aspects. Reviewer #1 recommended strengthening the structure and revise the discussion with relevant literature. Reviewer #2 did a thorough revision, and also suggest several changes in the title, interpretations, and recommend a revision of the language.

Reviewer 1 ·

Basic reporting

The paper represents comparison analysis of Copepod abundance and biomass in Gulf of Gdansk and in the more open sea area in front of this Gulf. It is mostly descriptive kind of study, however, includes valuable data of differences and similarities between biocenosis of coastal and open sea waters.
The paper should be improved in the structure and discussion with relevant literature. Also, the conclusion should be clarified. See detail suggestions in the "Validity of the findings":

Experimental design

The paper presents original results. The methods are described properly.

Validity of the findings

59-85– this chapter belongs to „Material and Methods”
68-69 Danish straits is an only connection between Baltic Sea and the North Sea.
112 – Should be a citation here.
116 – Part from “based…Gdańsk” is not necessary here.
119 – I propose to divide chapter “Material and methods” on two subchapters: “Study area” (text from lines 59-85 and also 367-383 with avoiding repetitions) and “Sampling” (current text in the chapter)
210 – “At station P1” is the only technical term. It should be changed to a geographical name.(the same: lines 238, 284, 303)
366-383 – It is not a discussion but the description of the study area. Should allocate to this (proposed) subchapter
384 -491 This part is the description of the results and belongs (after rejecting repetitive information) to the Results chapter.
492 – Subchapter should be supplemented by the comparisons of the Copepod abundance with data from other gulfs of Baltic with wider citation of relevant literature. Here, any discussion with literature is absent.
537-540 – Because Gulf of Gdansk is not isolated from the open sea, why it is unique? Comparing to what? The author declared in 536 line that unique (specific) biocenoses have Gulfs of Bothnia, Finnland, and Riga.

Additional comments

The paper should be improved in the structure and discussion with relevant literature. Also, the conclusion should be clarified.The paper needs revision according to detail remarks below, before publication in the Peerj.

Reviewer 2 ·

Basic reporting

For the paper to be publishable, the language needs to be strongly improved through much of the paper. At present some of the text seems somewhat premature, with issues regarding the wording and grammar, and the text needs to flow better. Improving the text, particularly in the Materials and Methods and Results, would make the paper easier to understand.

The literature references and field backgroud/context seems suffiecient.

Article structure, figures and tables are OK, and raw data are shared.

The submission is self-contained - and includes the relevant results.

This is an appropriate unit for publication

Experimental design

The paper is based on original primary research and is within the aims and scope of the journal.

The research question is well defined.

The methods used are adequate for adressing the research question.

In a few cases, noted specifically in the general comments to the authors, a little more detail should be given in the methodological desrption.

Validity of the findings

The submission is primarily descriptive - which is fine.

The study includes 2 sampling stations - which are located in different environments, Comparisons are made. Both stations are covered for almost two years and the within-station results for the two years are compared .

The results are considered to be robust.

The conclusions are well stated, based on the research questions, and puts these into a larger context.

Additional comments

Review of the manuscript:

«Seasonal changes in the abundance and biomass of copepod in the southwestern Baltic Sea in 2010 and 2011”

Lidia Dzierzbicka-Glowacka, Anna Lemieszek, Evelina Griniene, Marcin Kalarus

PeerJ

This paper is based on time-series on copepod species abundances and biomasses at two stations in the south-western Baltic Sea during 2010 and 2011. One station, P1 (Gdansk Deep), was located in a deeper area and considered to represent the open ocean region, while the other station, P2 (Gulf of Gdansk), was located closer to shore and mainly influenced by coastal water. The study covers the annual cycle over almost 2 years – at both stations. Plankton nets, Copenhagen net and WP2, were used for collection of samples. Both nets had mesh-size 100 µm, and were hauled vertically.
The focus of the paper is seasonal changes in abundances and biomasses of the main copepod species. The results focus on the copepods Acartia spp., Temora longicornis and Pseudocalanus sp., which were found to be the dominant zooplankton groups in this study, both with respect to numerical abundances and biomass.

The main results are the changes in copepod abundances and species-composition during the seasonal cycle, on a coastal versus open-ocean station, with a comparison of the results for 2 consequent years. The results are discussed in relation to hydrological variability – which was also measured. The paper is primarily descriptive, and provides basic yet essential ecosystem information.

Conclusion:
The data and results are interesting, relevant and valuable.

For the paper to be publishable, a major revision would be required. This is because the language needs to be strongly improved through much of the paper. At present some of the text seems somewhat premature, with issues regarding the wording and grammar, and the text needs to flow better. Improving the text, particularly in the Materials and Methods and Results, would make the paper easier to understand.

Regarding the presented concentrations of copepods and the species proportions, I would like to be assured that these (except in the cases when stated otherwise) actually represent the entire water-column as a whole, and that the calculations take into account that the sampling strata are not always equal regarding distance between upper and lower sampling depth. If this was not accounted for due to not being considered to be of importance, this should be clearly stated and explained. Further, it should be made clear which developmental stages that are included in the presented results – both in the text and figure legends.

Statistical analyses are not applied – but given the scope of the paper this would not seem necessary.

Specific comments:
In the title and throughout the paper the term “Copepod” is used. “Copepoda” is a taxonomic subclass within subphylum Crustacea, which belongs to phylum Arthropoda. Using the term “Copepod” with a capital “C” and without an “a” at the end, as in the title, does not seem to be correct. Why not simply use the term “copepods” (plural form) consistently? For instance, the title could then be “Seasonal changes in the abundance and biomass of copepods in the Baltic Sea in 2010 and 2011”. This modification could be made throughout the paper.

Summary page for the different chapters - on first page:
This is to the point, and generally well written.
I do find the geographical positions confusing due to the symbols phi and lambda (φ, λ), but this could be due to ignorance on my part. These symbols are used when describing geographical positions throughout the paper, both in text and figure-legends. Please make sure that the geographical system presented is right, and correct it throughout if it is not.
It is not clear to me what is meant by qualitative versus quantitative structure. Is this absence/presence versus relative and absolute abundances? This is written a couple of places in the paper.
Lines 24-56 in the paper: Same comments as above – as the text is very similar.
I will not mention all the cases where I consider the writing needs to be corrected or improved in the paper – but only give a few examples in the following.

Introduction
The contents generally seem alright, but there are issues regarding the English.
Some statements surprise me, but this could be due to the writing more than the intended message. An example is on lines 110-112: I hardly think that the physiochemical properties of the environment can be determined by absence-presence of different species (at least not with any precision) – but rather guess that the absence-presence of different species will reflect or indicate the physiochemical environment. I feel that the sentence now is too strong – perhaps more than intended?

Materials and methods
The language needs to be checked and improved in the whole chapter.
Line 120: “planktonic material”. Would it be better with “plankton samples”?
Line 124: Check the description of the geographic position. If this is correct, I apologize. This is relevant for the whole paper, including the figure-legends – but I won’t bring it up again.
Lines 126-127: It would be good to include some references for the plankton nets.
Line 131: “placed at 1/3 of the diameter of the net inlet”. Are you sure you do not mean “at 1/3 of the distance from the edge of the net towards its centre”? Please check and reformulate if necessary.
Line 138: Better with “ … a list of samples collected at station P1”?
Line 144: Better with “ … , from the ship of …”? Is the word “aboard” needed?
Lines 145-146: “The site of biological material collection was ..” I suggest changing to : “The location for collection of plankton samples …”
Line 146: I am not sure what Mm signifies. Is there an alternative way of giving the distance that is easier to understand – for instance in kilometres?
Line 148: “.. along the water column …”. How about “ throughout the water column”?
Line 151: “ … list of material at P2 station”. Would it be better with “ … list of samples collected at station P2”?
Line 154: Same comment as for line 131.
Line 156: Which developmental stages were identified, the copepodite stages and nauplii? A little more information should be given about this.
Figure 1: Some bottom-contours in the map would be informative for the readers.
The samples collected from both stations P1 and P2 are vertically stratified. My understanding here is based on the text as well as for instance the file “Data_S4.xlsx”. For both stations, concentrations of copepods are reported in the Results. Which depth-strata do these reported concentrations represent in those cases when no specific depth-stratum is mentioned? The whole water-column? If the results represent the entire water-column – how were the data for the different vertical layers combined? The depth layers are not always equally “wide” at station P2 (at least on some sampling occasions) – was this accounted for in the calculations? This is important information and should be given in a clear way in the Materials and methods – and in the Results (including the figure-legends) it should be specified which depth-stratum that the concentrations represent. I was not able to understand completely how the calculations were made by looking in the data-files – so this should be explained in the text of the Mat&Mat.

In the Results, salinity and temperature data are used to describe the environmental conditions at the time and location of plankton-sampling. The description of the collection of the salinity and temperature data should be given in the Materials and Methods – not the Results. I also think that the description of the CTD-probe(s) and how it/they was/were used should be a little more precise than what is presently given in the Results chapter.

Results:
The text should be checked thoroughly, and improved wherever possible.
Instead of listing so many numbers, I think it would be good to focus more on the main patterns and messages. In many cases it should be enough to refer to the figures for the details. That would probably make the chapter more readable.

Environmental conditions:
The description of the measurements should be moved to Materials and Methods, and only the results shown in the present chapter.
Line 186-187: Which increase is this? Please clarify.
Line 198: Which cases do “the differences in both cases” refer to? This is a bit confusing – please clarify.
Lines 199-200: “the temperature began to level off and uniformely fluctuated”. This is a bit hard to follow- can it be described in a way that is easier to understand?

Copepod abundance:
Please refer to the figures – that would ease the interpretation when reading the text.
In some places the text is hard to follow, for instance as in lines 224-227. As an example, I think that it there would be better to say that “Pseudocalanus elongatus had a higher relative proportion at station P1 than station P2”, and refer to figures 3 and 5. More “straight to the point”. Personally, I also think that the reason for this would belong better in the Discussion than the Results.
Which depth-strata do the zooplankton concentrations and species proportions represent? I assume it is the entire water-column. This must be specified in the text and in the figure legends.
Do the species abundances and proportions represent the sum of all copepodite stages? Again, this should be clarified.
Line 211: This seems obvious – I suggest removing it.
Line 214 and onwards: These concentrations (unless when specifically stating otherwise in the text) represent the pelagic zone, so I assume the concentrations then represent the average of the entire water column. Correctly understood?
Line 239: Also here, the first sentence seems rather obvious. Is it needed?
Line 245: “… over two times lower ..”. Better with “ … less than the half …”.
Line 249: I do not think this looks like two peaks (in 2011). There will always be some uncertainty associated with estimated abundances, and keeping that in mind, I rather think that the presented abundances look rather similar during May-July. If there is a peak in June-July, it is not easy to see in Fig. 4.
Line 281: Alternatively, you could say “In 2011, Eurytemora sp. was rare, only accounting for up to ….”.

Copepod biomass:
Line 312-313: Whether there are one or two peaks during June-September is not clear to me. What seems clear is that the biomass during these months are higher compared to the months before and after, but I fail to see two peaks. With my eyes, the biomasses seem rather similar during June-September.
Lines 314-315: Is this sentence needed? Perhaps just go straight to the nest sentence where things become more specific – at least that would be my choice.

Discussion:
The Discussion in general is very well written and sharp. This is the level to aim at for the whole paper.

Description of the study area:
This section flows nicely.
Perhaps the word “morphometric” in line 369 could be changed – I am unsure what it means in this context.

Taxonomic composition of plankton:
This section flows well, is interesting, and easy to follow. In the last part of the section, it would be nice to include some references, if the results from the previous studies mentioned are published.
Line 389: Worth mentioning that Hyperia galba is a hyperiid or amphipod?
Lines 390-392: “The percentage of individual taxa and their vertical distribution were determined by metereological and hydrological conditions prevailing at a given time”. This is a rather strong statement, even if it may very well be true. Still, as no statistical testing of the relationships between environmental variables and the plankton species composition and their distributions were made in this study, I would moderate this statement. If the statement refers to other research or general knowledge – it should be cited.

Changes in the abundance and biomass of zooplankton:
Good.


Conclusion
It ties up the results and puts them in a larger context, while also explaining why the results of the study are valuable and how the can be used in the future.

---

## Round 0.2 · Minor Revisions

The authors have resolved the major issues pointed by the reviewer, and the revised version is much improved, satisfying the reviewer and editor concerns.

# Reviewer 1 ·

Basic reporting

The authors improved the article according to my expectations.

Experimental design

No comments.

Validity of the findings

No comments.

Additional comments

The answers of the authors to my review are satisfied and changes in the text follow my suggestions. From my point of view, the manuscript in the present form can be considered for publication.

---

## Round 0.3 · accepted · Accept

The article is now Accepted.

However, a staff check found several issues remain in the English language. Please work with the production team to do a final round of language editing.